# Discovering Hidden Variables in Noisy-Or Networks using Quartet Tests

**Yacine Jernite,   Yoni Halpern,   David Sontag**
Courant Institute of Mathematical Sciences
New York University
{halpern, jernite, dsontag}@cs.nyu.edu

## Abstract

We give a polynomial-time algorithm for provably learning the structure and parameters of bipartite noisy-or Bayesian networks of binary variables where the top layer is completely hidden. Unsupervised learning of these models is a form of discrete factor analysis, enabling the discovery of hidden variables and their causal relationships with observed data. We obtain an efficient learning algorithm for a family of Bayesian networks that we call quartet-learnable. For each latent variable, the existence of a singly-coupled quartet allows us to uniquely identify and learn all parameters involving that latent variable. We give a proof of the polynomial sample complexity of our learning algorithm, and experimentally compare it to variational EM.

## 1   Introduction

We study the problem of discovering the presence of latent variables in data and learning models involving them. The particular family of probabilistic models that we consider are bipartite noisy-or Bayesian networks where the top layer is completely hidden. Unsupervised learning of these models is a form of discrete factor analysis and has applications in sociology, psychology, epidemiology, economics, and other areas of scientific inquiry that need to identify the causal relationships of hidden or latent variables with observed data (Saund, 1995; Martin & VanLehn, 1995). Furthermore, these models are widely used in expert systems, such as the QMR-DT network for medical diagnosis (Shwe *et al.* , 1991). The ability to learn the structure and parameters of these models from partially labeled data could dramatically increase their adoption.

We obtain an efficient learning algorithm for a family of Bayesian networks that we call quartet-learnable, meaning that every latent variable has a *singly-coupled quartet* (i.e. four children of a latent variable for which there is no other latent variable that is shared by at least two of the children). We show that the existence of such a quartet allows us to uniquely identify each latent variable and to learn all parameters involving that latent variable. Furthermore, using a technique introduced by Halpern & Sontag (2013), we show how to *subtract* already learned latent variables to create new singly-coupled quartets, substantially expanding the class of structures that we can learn. Importantly, even if we cannot discover every latent variable, our algorithm guarantees the correctness of any latent variable that was discovered. We show in Sec. 4 that our algorithm can learn nearly all of the structure of the QMR-DT network for medical diagnosis (i.e., discovering the existence of hundreds of diseases) simply from data recording the symptoms of each patient.

Underlying our algorithm are two new techniques for structure learning. First, we introduce a quartet test to determine whether a set of binary variables is singly-coupled. When singly-coupled variables are found, we use previous results in mixture model learning to identify the coupling latent variable. Second, we develop a conditional point-wise mutual information test to learn parameters of other children of identified latent variables. We give a self-contained proof of the polynomial sample

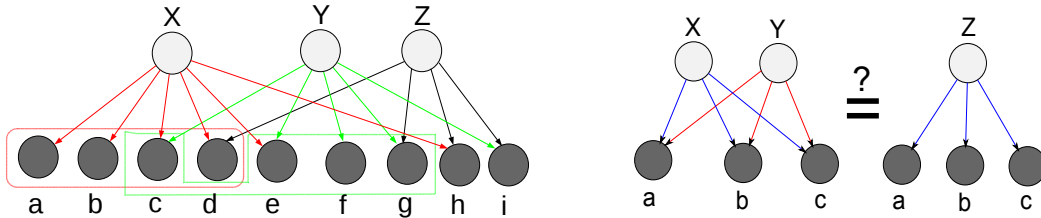

Figure 1: **Left:** Example of a quartet-learnable network. For this network, the order $(X, Y, Z)$ satisfies the definition: $\{a, b, c, d\}$ is singly coupled by $X$, $\{c, e, f, g\}$ is singly coupled by $Y$ given $X$ and $\{d, g, h, i\}$ is singly coupled by $Z$ given $X, Y$. **Right:** Example of two different networks that have the same observable moments (i.e., distribution on $a, b, c$). $p_X = 0.2$, $p_Y = 0.3$, $p_Z = 0.37$. $f_X = (0.1, 0.2, 0.3)$, $f_Y = (0.6, 0.4, 0.5)$, $f_Z = (0.28, 0.23, 0.33)$. The noise probabilities and full moments are given in the supplementary material.

complexity of our structure and parameter learning algorithms, by bounding the error propagation due to finding roots of polynomials. Finally, we present an experimental comparison of our structure learning algorithm to the variational expectation maximization algorithm of Šingliar & Hauskrecht (2006) on a synthetic image-decomposition problem and show competitive results.

**Related work.** Martin & VanLehn (1995) study structure learning for noisy-or Bayesian networks, observing that any two observed variables that share a hidden parent must be correlated. Their algorithm greedily attempts to find a small set of cliques that cover the dependencies of which it is most certain. Kearns & Mansour (1998) give a polynomial-time algorithm with provable guarantees for structure learning of noisy-or bipartite networks with bounded in-degree. Their algorithm incrementally constructs the network, in each step adding a new observed variable, introducing edges from the existing latent variables to the observed variable, and then seeing if new latent variables should be created. This approach requires strong assumptions, such as identical priors for the hidden variables and all incoming edges for an observed variable having the same failure probabilities.

Silva *et al.* (2006) study structure learning in linear models with continuous latent variables, giving an algorithm for discovering disjoint subsets of observed variables that have a single hidden variable as its parent. Recent work has used tensor methods and sparse recovery to learn linear latent variable models with graph expansion (Anandkumar *et al.* , 2013), and also continuous admixture models such as latent Dirichlet allocation (Anandkumar *et al.* , 2012a). The discrete variable setting is not linear, making it non-trivial to apply these methods that rely on linearity of expectation. An alternative approach is to perform gradient ascent on the likelihood or use expectation maximization (EM). Although more robust to model error, the likelihood is nonconvex and these methods do not have consistency guarantees. Elidan *et al.* (2001) seek "structural signatures", in their case semicliques, to use as structure candidates within structural EM (Elidan & Friedman, 2006; Friedman, 1997; Lazic *et al.* , 2013). Our algorithm could be used in the same way.

Exact inference is intractable in noisy-or networks (Cooper, 1987), so Šingliar & Hauskrecht (2006) give a variational EM algorithm for unsupervised learning of the parameters of a bipartite noisy-or network. We will use this as a baseline in our experimental results.

Spectral approaches to learning mixture models originated with Chang's spectral method (Chang 1996; analyzed in Mossel & Roch 2005, see also Anandkumar *et al.* (2012b)). The binary variable setting is a special case and is discussed in Lazarsfeld (1950) and Pearl & Tarsi (1986). In Halpern & Sontag (2013) the parameters of singly-coupled variables in bipartite networks of known structure are learned using mixture model learning.

Quartet tests have been previously used for learning latent tree models (Anandkumar *et al.* , 2011; Pearl & Tarsi, 1986). Our quartet test, like that of Ishteva *et al.* (2013) and Eriksson (2005), uses the full fourth-order moment and a similar unfolding of the fourth-order moment matrix.

**Background**. We consider bipartite noisy-or Bayesian networks $(\mathcal{G}, \Theta)$ with $n$ binary latent variables $\mathcal{U}$, which we denote with capital letters (e.g. $X$), and $m$ observed binary variables $\mathcal{O}$, which we denote with lower case letters (e.g. $a$). The edges in the model are directed from the latent variables to the observed variables, as shown in Fig. 1. In the noisy-or framework, an observed variable is on if at least one of its parents is on and does not fail to activate it.

The entire Bayesian network is parametrized by $n \times m + n + m$ parameters. These parameters consist of *prior* probabilities on the latent variables, $p_X$ for $X \in \mathcal{U}$, *failure* probabilities between latent and

observed variables, $\vec{f}_X$ (a vector of size $m$), and noise or leak probabilities $\vec{\nu} = \{\nu_1, ..., \nu_m\}$. An equivalent formulation includes the noise in the model by introducing a single 'noise' latent variable, $X_0$, which is present with probability $p_0 = 1$ and has failure probabilities $\vec{f}_0 = 1 - \vec{\nu}$. The Bayesian network only has an edge between latent variable $X$ and observed variable $a$ if $f_{X,a} < 1$. The generative process for the model is then:

- The states of the latent variables are drawn independently: $X \sim \text{Bernoulli}(p_X)$ for $X \in \mathcal{U}$.
- Each $X \in \mathcal{U}$ with $X = 1$ activates observed variable $a$ with probability $1 - f_{X,a}$.
- An observed variable $a \in \mathcal{O}$ is "on" ($a = 1$) if it is activated by at least one of its parents.

The algorithms described in this paper make substantial use of sets of moments of the observed variables, particularly the *negative* moments. Let $\mathcal{S} \subseteq \mathcal{O}$ be a set of observed variables, and $\mathcal{X} \subseteq \mathcal{U}$ be the set of parents of $\mathcal{S}$. The joint distribution of a bipartite noisy-or network can be shown to have the following factorization, where $\mathcal{S} = \{o_1, \ldots, o_{|\mathcal{S}|}\}$:

$$N_{\mathcal{G},\mathcal{S}} = P(o_1 = 0, o_2 = 0, \ldots, o_{|\mathcal{S}|} = 0) = \prod_{U \in \mathcal{X}} \left(1 - p_U + p_U \prod_{i=1}^{|\mathcal{S}|} f_{U,o_i}\right). \quad (1)$$

The full joint distribution can be obtained from the negative moments via inclusion-exclusion formulas. We denote $N_\mathcal{G}$ to be the set of negative moments of the observed variables under $(\mathcal{G}, \Theta)$. In the remainder of this section we will review two results described in Halpern & Sontag (2013).

**Parameter learning of singly-coupled triplets**. We say that a set $\mathcal{O}$ of observed variables is singly-coupled by a parent $X$ if $X$ is a parent of every member of $\mathcal{O}$ and there is no other parent $Y$ that is shared by at least two members of $\mathcal{O}$. A singly coupled set of observations is a binary mixture model, which gives rise to the next result based on a rank-2 tensor decomposition of the joint distribution.

If $(a, b, c)$ are singly-coupled by $X$, we can learn $p_X$ and $f_{X,a}$ as follows. Let $M_1 = P(b, c, a = 0)$, $M_2 = P(b, c, a = 1)$, and $M_3 = M_2 M_1^{-1}$. Solving for $(\lambda_1, \lambda_2) = \text{eigenvalues}(M_3)$, we then have:

$$p_X = \frac{1 + \lambda_2}{\lambda_2 - \lambda_1} \mathbf{1}^T (M_2 - \lambda_1 M_1) \mathbf{1} \quad \text{and} \quad f_{X,a} = \frac{1 + \lambda_1}{1 + \lambda_2}. \quad (2)$$

**Subtracting off.** Because of the factored form of Equation 1, we can remove the influence of a latent variable from the negative moments. Let $X$ be a latent variable of $\mathcal{G}$. Let $\mathcal{S} \subseteq \mathcal{O}$ be a set of observations and $\mathcal{X}$ be the parents of $\mathcal{S}$. If we know $N_{\mathcal{G},\mathcal{S}}$, the prior of $X$, and the failure probabilities $f_{X,\mathcal{S}}$, we can obtain the negative moments of $\mathcal{S}$ under $(\mathcal{G} \setminus \{X\}, \Theta)$. When $\mathcal{S}$ includes all of the children of $X$, this operation "subtracts off" or removes $X$ from the network:

$$N_{\mathcal{G} \setminus X, \mathcal{S}} = \prod_{U \in \mathcal{X} \setminus X} \left(1 - p_U + p_U \prod_{i=1}^{|\mathcal{S}|} f_{U,o_i}\right) = \frac{N_{\mathcal{G},\mathcal{S}}}{\left(1 - p_X + p_X \prod_{i=1}^{|\mathcal{S}|} f_{X,o_i}\right)}. \quad (3)$$

## 2 Structure learning

Our paper focuses on learning the structure of these bipartite networks, including the number of latent variables. We begin with the observation that not all structures are identifiable, even if given infinite data. Suppose we applied the tensor decomposition method to the marginal distribution (moments) of three observed variables that share two parents. Often we can learn a network with the same marginal distribution, but where these three variables have just one parent. Figure 1 gives an example of such a network. As a result, if we hope to be able to learn structure, we need to make additional assumptions (e.g., every latent variable has at least four children).

We give two variants of an algorithm based on quartet tests, and prove its correctness in Section 3. Our approach is based on decomposing the structure learning problem into two tasks: (1) identifying the latent variables, and (2) determining to which observed variables they are connected.

### 2.1 Finding singly coupled quartets

Since triplets are not sufficient to identify a latent variable (Figure 1), we propose a new approach based on identifying singly-coupled quartets. We present two methods to find such quartets. The

**Algorithm 1** STRUCTURE-LEARN

**Input:** Observations $\mathcal{S}$, Thresholds $\tau_q$, $\tau_q'$, $\tau_e$.
**Output:** Latent structure $Latent$
1: $Latent = \{\}$
2: **while** Not Converged **do**
3:   **for all** quartets $(a, b, c, d)$ in $\mathcal{S}$ **do**
4:     $T \leftarrow \text{JOINT}(a, b, c, d)$
5:     $T \leftarrow \text{ADJUST}(T, Latent)$
6:     **if** PRETEST($T$,$\tau_e$) **and** 4TEST($T$, $\tau_q$,$\tau_q'$) **then**
7:       *// $(a, b, c, d)$ are singly-coupled.*
8:       $L \leftarrow \text{MIXTURE}(a, b, c, d)$
9:       $children \leftarrow \text{EXTEND}(L, Latent, \tau_e)$
10:       $Latent \leftarrow Latent \cup \{(L, children)\}$
11:     **end if**
12:   **end for**
13: **end while**

**Algorithm 2** EXTEND

**Input:** Latent variable $L$ with singly-coupled children $(a, b, c, d)$, currently known latent structure $Latent$, threshold $\tau$
**Output:** $children$, all the children of $L$.
1: $children = \{(a, f_{L,a}), (b, f_{L,b}),$ $(c, f_{L,c}), (d, f_{L,d})\}$
2: **for all** observable $x \notin \{a, b, c, d\}$ **do**
3:   Subtract off coupling parents in $Latent$ from the moments
4:   **if** $\frac{P(\bar{a},\bar{b})}{P(\bar{a})P(\bar{b})} > \frac{P(\bar{a},\bar{b}|\bar{x})}{P(\bar{a}|\bar{x})P(\bar{b}|\bar{x})} + \tau$ **then**
5:     $f_{L,x} = \text{FAILURE}(a,b,x,L)$
6:     $children \leftarrow children \cup \{(x, f_{L,x})\}$
7:   **end if**
8: **end for**
9: Return $children$

Figure 2: Structure learning. **Left:** Main routine of the algorithm. JOINT gives the joint distribution and ADJUST subtracts off the influence of the latent variables (Eq. 3). PRETEST filters the set of candidate quartets by determining whether every triplet in a quartet has a shared parent, using Lemma 2. 4TEST refers to either of the quartet tests described in Section 2.1. $\tau_q'$ is only used in the coherence quartet test. MIXTURE refers to using Eq. 2 to learn the parameters for all triplets in a singly-coupled quartet. This yields multiple estimates for each parameter and we take the median. **Right:** Algorithm to identify all of the children of a latent variable. FAILURE uses the method outlined in Section 2.2 (see Eq. 6) to find the failure probability $f_{L,x}$.

first is based on a rank test on a matrix formed from the fourth order moments and the second uses variance of parameters learned from third order moments. We then present a method that uses the point-wise mutual information of a triplet to identify all the other children of the new latent variable. The outline of the learning algorithm is presented in Algorithm 1.

While not all networks can be learned, this method allows us to define a class of noisy-or networks on which we can perform structure learning.

**Definition 1.** *A noisy-or network is quartet-learnable if there exists an ordering of its latent variables such that each one has a quartet of children which are singly coupled once the previous latent variables are removed from the model. A noisy-or network is **strongly** quartet-learnable if all of its latent variables have a singly coupled quartet of children.*

An example of a quartet-learnable network is given in Figure 1.

**Rank test**. A candidate quartet for the rank test is a quartet where all nodes have at least one common parent. One way to find whether a candidate quartet is singly coupled is by looking directly at the rank of its fourth-order moments matrix. We have three ways to unfold the $2 \times 2 \times 2 \times 2$ tensor defined by these moments into a $4 \times 4$ matrix: we can consider the joint probability matrix of the aggregated variables $(a, b)$ and $(c, d)$, of $(a, c)$ and $(b, d)$, or of $(a, d)$ and $(b, c)$. We discuss the rank property for the first unfolding, but note that it holds for all three.

Let $M$ be the $4 \times 4$ matrix obtained this way, and $\mathcal{S}$ be the set of parents that are parents of both $(a, b)$ and $(c, d)$. For all $S \subset \mathcal{S}$ let $q_S$ and $r_S$ be the vectors of the probabilities of $(a, b)$ and $(c, d)$ respectively given that $S$ is the set of parents that are active. Then:

$$M = \sum_{S \subset \mathcal{S}} \left( \prod_{X \in S} p_X \prod_{Y \in \mathcal{S} \setminus S} (1 - p_Y) \right) q_S r_S^T.$$

In particular, this means that if there is only one parent shared between $(a, b)$ and $(c, d)$, $M$ is the sum of two rank 1 matrices, and thus is at most rank 2.

Conversely, if $|\mathcal{S}| > 1$, $M$ is the sum of at least 4 rank 1 matrices, and its elements are polynomial expressions of the parameters of the model. The determinant itself is then a polynomial function of the parameters of the model, i.e. $P(p_X, f_{X,u} \,\forall X \in \mathcal{S}, u \in \{a, b, c, d\})$. We give examples in the supplementary material of parameter settings showing that $P \not\equiv 0$, hence the set of its roots has measure 0, which means that the third largest eigenvalue (using the eigenvalues' absolute values) of $M$ is non-zero with probability one.

This will allow us to determine whether a candidate quartet is singly coupled by looking at the third eigenvalues of the three unfoldings of its joint distribution tensor. However, for the algorithm to be practical, we need a slightly stronger formalization of the property:

**Definition 2.** *We say that a model is $\epsilon$-rank-testable if for any quartet $\{a, b, c, d\}$ that share a parent $U$ and any non-empty set of latent variables $\mathcal{V}$ such that $U \notin \mathcal{V}$ and $\exists V \in \mathcal{V}, (f_{V,b} \neq 1 \wedge f_{V,c} \neq 1)$, the third eigenvalue of the moments matrix M corresponding to the sub-network $\{U, a, b, c, d\} \cup \mathcal{V}$ is at least $\epsilon$.*

Any (finite) noisy-or network whose parameters were drawn at random is $\epsilon$-rank-testable for some $\epsilon$ with probability 1. The special case where all failure probabilities are equal also falls within this framework, provided they are not too close to 0 or 1. We can then determine whether a quartet is singly coupled by testing whether the third eigenvalues of all of the three unfoldings of the joint distributions are below a threshold, $\tau_q$. If this test succeeds, we learn its parameters using Eq. 2.

**Coherence test**. Let $\{a, b, c, d\}$ be a quartet of observed variables. To determine whether it is singly coupled, we can also apply Eq. 2 to learn the parameters of triplets $(a, b, c)$, $(a, b, d)$, $(a, c, d)$ and $(b, c, d)$ as if they were singly coupled. This gives us four overlapping sets of parameters. If the variance of parameter estimates exceeds a threshold we know that the quartet is not singly coupled.

Note that agreement between the parameters learned is necessary but not sufficient to determine that $(a, b, c, d)$ are singly coupled. For example, in the case of a fully connected graph with two parents, four children and identical failure probabilities, the third-order moments of any triplet are identical, hence the parameters learned will be the same. Lemma 1, however, states that the moments generated from the estimated parameters can only be equal to the true moments if the quartet is actually singly coupled.

**Lemma 1.** *If the model is $\epsilon$-rank-testable and $(a, b, c, d)$ are not singly coupled, then if $M_R$ represents the reconstructed moments and $M$ the true moments, we have:*

$$||M_R - M||_\infty > \left(\frac{\epsilon}{8}\right)^4.$$

This can be proved using a result on eigenvalue perturbation from Elsner (1985) for an unfolding of the moments' tensor. These two properties lead to the following algorithm: First try to learn the parameters as if the quartet were singly coupled. If the variance of the parameter estimates exceeds a threshold, then reject the quartet. Next, check whether we can reconstruct the moments using the mean of the parameter estimates. Accept the quartet as singly-coupled if the reconstruction error is below a second threshold.

## 2.2 Extending Latent Variables

Once we have found a singly coupled quartet $(a, b, c, d)$, the second step is to find all other children of the coupling parent $A$. To that end, we can use a property of the conditional point-wise mutual information (CPMI) that we introduce in this section. In this section, we use the notation $\bar{a}$ to denote the event $a = 0$. The CPMI between $a$ and $b$ given $x$ is defined as $\text{CPMI}(a, b|x) \equiv P(\bar{a}, \bar{b}|\bar{x})/(P(\bar{a}|\bar{x})P(\bar{b}|\bar{x}))$. We will compare it to the point-wise mutual information (PMI) between $a$ and $b$ defined as $\text{PMI}(a, b) \equiv P(\bar{a}, \bar{b})/(P(\bar{a})P(\bar{b}))$.

Let $(a, b)$ be two observed variables that we know only share one parent $A$, and let $x$ be any another observed variable. We show how the CPMI between $a$ and $b$ given $x$ can be used to find $f_{A,x}$, the failure probability of $x$ given $A$. Our algorithm requires that the priors of all of the hidden variables be less than 1/2.

For any observed variable $x$, the following lemma states that $\text{CPMI}(a, b|x) \neq \text{PMI}(a, b)$ if and only if $a$, $b$ and $x$ share a parent. Since the only latent variable that has both $a$ and $b$ as children is $A$, this is equivalent to saying that $x$ is a child of $A$.

**Lemma 2.** *Let $(a, b, x)$ be three observed variables in a noisy-or network, and let $\mathcal{U}_{a,b}$ be the set of common parents of $a$ and $b$. For $U \in \mathcal{U}_{a,b}$, defining*

$$p_{U|\bar{x}} = \frac{P(U, \bar{x})}{P(\bar{x})} = \frac{p_U f_{U,x}}{1 - p_U + p_U f_{U,x}},\qquad(4)$$

*we have $p_{U|\bar{x}} \leq p_U$. Furthermore,*

$$\frac{P(\bar{a}, \bar{b}|\bar{x})}{P(\bar{a}|\bar{x})P(\bar{b}|\bar{x})} = \prod_{U \in \mathcal{U}_{a,b}} \frac{(1 - p_{U|\bar{x}} + p_{U|\bar{x}} f_{U,a} f_{U,b})}{(1 - p_{U|\bar{x}} + p_{U|\bar{x}} f_{U,a})(1 - p_{U|\bar{x}} + p_{U|\bar{x}} f_{U,b})} \leq \frac{P(\bar{a}, \bar{b})}{P(\bar{a})P(\bar{b})},$$

*with equality if and only if $(a, b, x)$ do not share a parent.*

The proof for Lemma 2 is given in the supplementary material. As a result, if $a$ and $b$ have only parent $A$ in common, we can write:

$$R \equiv \mathrm{CPMI}(a, b|x) = \frac{P(\bar{a}, \bar{b}|\bar{x})}{P(\bar{a}|\bar{x})P(\bar{b}|\bar{x})} = \frac{(1 - p_{A|\bar{x}} + p_{A|\bar{x}} f_{A,a} f_{A,b})}{(1 - p_{A|\bar{x}} + p_{A|\bar{x}} f_{A,a})(1 - p_{A|\bar{x}} + p_{A|\bar{x}} f_{A,b})}.$$

We can equivalently write this equation as $Q(p_{A|\bar{x}}) = 0$ for the quadratic function $Q(x)$ given by:

$$Q(x) = R(f_{A,a} - 1)(f_{A,b} - 1)x^2 + [R(f_{A,a} + f_{A,b} - 2) - (f_{A,a}f_{A,b} - 1)]x + R - 1. \quad(5)$$

Moreover, we can show that $Q'(x) = 0$ for some $x > 1/2$, hence one of the roots of Q is always greater than $1/2$. In our framework, we know that $p_{A|\bar{x}} \leq p_A \leq \frac{1}{2}$, hence $p_{A|\bar{x}}$ is simply the smaller root of $Q$. After solving for $p_{A|\bar{x}}$, we can obtain $f_{A,x}$ using Eq. 4:

$$f_{A,x} = \frac{p_{A|\bar{x}}(1 - p_A)}{p_A(1 - p_{A|\bar{x}})}.\qquad(6)$$

**Extending step.** Once we find a singly-coupled quartet $(a, b, c, d)$ with common parent $A$, Lemma 2 allows us to determine whether a new variable $x$ is also a child of $A$. Notice that for this step we only need to use two of the children in $\{a, b, c, d\}$, which we arbitrarily choose to be $a$ and $b$. If $x$ is found to be a child of $A$, we can solve for $f_{A,x}$ using Eqs. 5 and 6. Algorithm 2 combines these two steps to find the parameters of all the children of $A$ after a singly-coupled quartet has been found.

**Parameter learning with known structure.** When the structure of the network is known, singly-coupled triplets are sufficient for identifiability without resorting to the quartet tests in Section 2.1. That setting was previously studied in Halpern & Sontag (2013), which required *every edge* to be part of a singly coupled triplet or pair for its parameters to be learnable (possibly after subtracting off latent variables). Our new CPMI technique improves this result by enabling us to learn all failure probabilities for a latent variable's children even if the variable has only *one* singly coupled triplet.

## 3 Sample complexity analysis

In Section 2, we gave two variants of an algorithm to learn the structure of a class of noisy-or networks. We now want to upper bound the number of samples it requires to learn the structure of the network correctly with high probability, as a function of the ranges in which the parameters are found. All priors are in $[p_{min}, 1/2]$, all failures probabilities are in $[f_{min}, f_{max}]$, and the marginal probabilities of an observed variable $x$ being off is lower bounded by $n_{min} \leq P(\bar{x})$. The full proofs for these results are given in the supplementary materials.

**Theorem 1.** *If a network with $m$ observed variables is strongly quartet-learnable and $\zeta$-rank-testable, then its structure can be learned in polynomial time with probability $(1 - \delta)$ and with a polynomial number of samples equal to:*

$$O\Big( \max\Big( \frac{1}{\zeta^8}, \frac{1}{n_{min}^8 p_{min}^2 (1 - f_{max})^8} \Big) \ln\Big( \frac{2m}{\delta} \Big) \Big).$$

*After $N$ samples, the additive error on any of the parameters $\epsilon(N)$ is bounded with probability $1 - \delta$ by:*

$$\epsilon(N) \leq O\Big( \frac{\sqrt{\ln\big( \frac{2m}{\delta} \big)}}{f_{min}^{18}(1 - f_{max})^6 n_{min}^{28} p_{min}^{13}} \frac{1}{\sqrt{N}} \Big).$$

We obtain this result by determining the accuracy we need for our tests to be provably correct, and bounding how much the error in the output of the parameter learning algorithms depends on the input. This proves that we can learn a class of **strongly** quartet-learnable noisy-or networks in polynomial time and sample complexity. Next, we show how to extend the analysis to quartet-learnable networks as defined in Section 2 by subtracting off latent variables that we have previously learned. If some of the removed latent variables were coupling for an otherwise singly coupled quartet, we then discover new latent variables, and repeat the operation. If a network is quartet-learnable, we can find all of the latent variables in a finite number of subtracting off steps, which we call the **depth** of the network (thus, a strongly quartet-learnable network has depth 0). To prove that the structure learning algorithm remains correct, we simply need to show that the estimated subtracted off moments remain close to the true ones.

**Lemma 3.** *If the additive error on the estimated negative moments of an observed quartet $\mathcal{C}$ and on the parameters for $W$ latent variables $X_1, \ldots, X_W$ whose influence we want to remove from $\mathcal{C}$ is at most $\epsilon$, then the error on the subtracted off moments for $\mathcal{C}$ is $O(W 4^W \epsilon)$.*

We define the **width** of the network to be the maximum number of parents that need to be subtracted off to be able to learn the parameters for a new singly-coupled quartet (this is typically a small constant). This leads to the following result:

**Theorem 2.** *If a network with $m$ observed variables is quartet-learnable at depth d, is $\zeta$-rank-testable, and has width $W$, then its structure can be learned with probability $(1 - \delta)$ with $N_S$ samples, where:*

$$N_S = O\Big(\Big(\frac{W 4^W}{f_{min}^{18}(1 - f_{max})^6 n_{min}^{28} p_{min}^{13}}\Big)^{2d} \times \max\Big(\frac{1}{\zeta^8}, \frac{1}{n_{min}^8 p_{min}^2 (1 - f_{max})^8}\Big) \ln\Big(\frac{2m}{\delta}\Big)\Big).$$

The left hand side of this expression has to do with the error introduced in the estimate of the parameters each time we do a subtracting off step, which by definition occurs at most $d$ times, hence the exponent. We notice that the bounds do not depend directly on the number of latent variables, indicating that we can learn networks with many latent variables, as long as the number of subtraction steps is small. While this bound is useful for proving that the sample complexity is indeed polynomial, in the experiments section we show that in practice our algorithm obtains reasonable results on sample sizes well below the theoretical bound.

## 4  Experiments

**Depth of aQMR-DT**. Halpern & Sontag (2013) previously showed that the parameters of the anonymized QMR-DT network for medical diagnosis (provided by the University of Pittsburgh through the efforts of Frances Connell, Randolph A. Miller, and Gregory F. Cooper) could be learned from data recording only symptoms if the structure is known. We now show that the structure can also be learned. Here we assume that the quartet tests are perfect (i.e. infinite data setting). Table 1 compares the depth of the aQMR-DT network using triplets and quartets. Structure learning discovers all but four of the diseases, two of which would not be learnable even if the structure were known. These two diseases are discussed in Halpern & Sontag (2013) and share all of their children except for one symptom each, resulting in a situation where no singly-coupled triplets can be found. The additional two diseases that cannot be learned share all but two children with each other. Thus, for these two latent variables, singly-coupled triplets exist but singly-coupled quartets do not.

**Implementation**. We test the performance of our algorithm on the synthetic image dataset used in Šingliar & Hauskrecht (2006). The Bayesian network consists of 8 latent variables and 64 observed variables, arranged in an 8x8 grid of pixels. Each of the latent variables connects to a subset of the observed pixels (see Figure 3). The latent variable priors are set to 0.25, the failure probabilities for all edges are set to 0.1, and leak probabilities are set to 0.001. We generate samples from the network and use them to test the ability of our algorithm to discover the latent variables and network structure from the samples. The network is quartet learnable, but the first and last of the ground truth sources shown in Figure 3 can only be learned after a subtraction step.

We use variational EM (Šingliar & Hauskrecht, 2006) as a baseline, using 16 random initializations and choosing the run with the highest lower bound on likelihood. We found that multiple initializations substantially improved the quality of its result. The variational algorithm is given the correct

| Triplets (known structure) | | | Quartets (unknown structure) | | |
|---|---|---|---|---|---|
| depth | priors learned | edges learned | depth | diseases discovered | edges learned |
| 0 | 527 | 43,139 | 0 | 469 | 39,522 |
| 1 | 39 | 2,109 | 1 | 82 | 4,875 |
| 2 | 2 | 100 | 2 | 13 | 789 |
| 3 | 0 | 0 | 3 | 2 | 86 |
| inf | 2 | 122 | inf | 4 | 198 |

Table 1: **Right:** The depth at which latent variables (i.e., diseases) are discovered and parameters learned in the aQMR-DT network for medical diagnosis (Shwe *et al.* , 1991) using the quartet-based structure learning algorithm, assuming infinite data. **Left:** Comparison to parameter learning with known structure, using one singly-coupled triplet to learn the failure probabilities for all of a disease's symptoms. The parameters learned at level 0 can be learned without any subtracting-off step. Those marked depth inf cannot be learned.

number of sources as input. For our algorithm, we use the rank-based quartet test, which has the advantage of requiring only one threshold, $\tau_q$, compared to the two needed by the coherence test. In our algorithm, the thresholds determine the number of discovered latent variables (sources).

Quartets are pre-filtered using pointwise mutual information to reject quartets that have non-siblings (i.e. $(a, b, c, d)$ where $a$ and $b$ are likely not siblings). All quartets that fail the pretest or the rank test are discarded. We sort the remaining quartets by third singular value and proceed from lowest to highest. For each quartet in sorted order we check if it overlaps with a latent variable previously learned in this round. If it does not, we create a new latent variable and use the EXTEND step to find all of its children. The algorithm converges when no quartets pass the threshold.

Figure 3 shows how the algorithms perform on the synthetic dataset with varying numbers of samples. Unless otherwise specified, our experiments use threshold values $\tau_q = 0.01$ and $\tau_e = 0.1$. Experiments exploring the sensitivity of the algorithm to these thresholds can be found in the supplementary material. The running time of the quartet algorithm is under 6 minutes for 10,000 samples using a parallel implementation with 16 cores. For comparison, the variational algorithm on the same samples takes 4 hours using 16 cores simultaneously (one random initialization per core) on the same machine. The variational run-time scales linearly with sample size while the quartet algorithm is independent of sample size once the quartet marginals are computed.

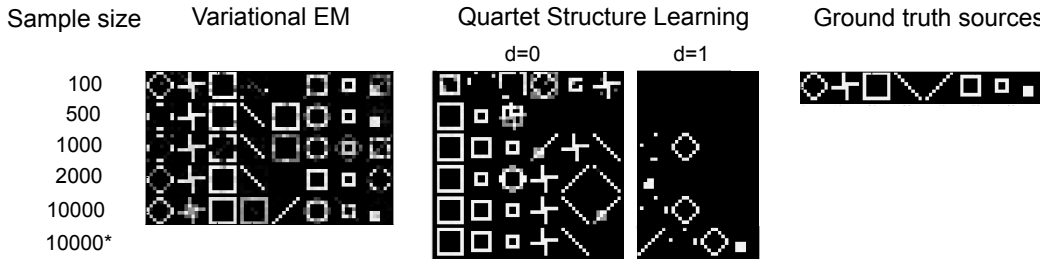

Figure 3: A comparison between the variational algorithm of Šingliar & Hauskrecht (2006) and the quartet algorithm as the number of samples increases. The true network structure is shown on the right, with one image for each of the eight latent variables (sources). For each edge from a latent variable to an observed variable, the corresponding pixel intensity specifies $1 - f_{X,a}$ (black means no edge). The results of the quartet algorithm are divided by depth. Column d=0 shows the sources learned without any subtraction and d=1 shows the sources learned after a single subtraction step. Nothing was learned at $d > 1$. The sample size of 10,000* refers to 10,000 samples using an optimized value for the threshold of the rank-based quartet test ($\tau_q = 0.003$).

## 5   Conclusion

We presented a novel algorithm for learning the structure and parameters of bipartite noisy-or Bayesian networks where the top layer consists completely of latent variables. Our algorithm can learn a broad class of models that may be useful for factor analysis and unsupervised learning. The structure learning algorithm does not depend on an ability to estimate the parameters in strongly quartet-learnable networks. As a result, it may be possible to generalize the approach beyond the noisy-or setting to other bipartite Bayesian networks, including those with continuous variables and discrete variables of more than two states.

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
