[Supplementary Material]

# Discovering Hidden Variables in Noisy-Or Networks using Quartet Tests – Supplementary Material

**Yacine Jernite,   Yoni Halpern,   David Sontag**
Courant Institute of Mathematical Sciences
New York University
{halpern, jernite, dsontag}@cs.nyu.edu

## A   Quartet tests pseudo-code

The pseudo-code for the rank and coherence tests described in Section 2.1 are presented below in Algorithms 1 and 2:

---
**Algorithm 1** QUARTET:Rank test
---
**Require:** quartet $(a, b, c, d)$, empirical moments $\hat{M}_{(a,b,c,d)}$, threshold $\tau_q$
**Ensure:** boolean singly-coupled.
 1: $M_{(a,b)} = \text{unfold}(\hat{M}_{(a,b,c,d)}, (a, b), (c, d))$
 2: $M_{(a,c)} = \text{unfold}(\hat{M}_{(a,b,c,d)}, (a, c), (b, d))$
 3: $M_{(a,d)} = \text{unfold}(\hat{M}_{(a,b,c,d)}, (a, d), (b, c))$
 4: **for** $u \in \{b, c, d\}$ **do**
 5:     $\lambda_{3,(a,u)} \leftarrow$ third eigenvalue of $M_{(a,u)}$
 6: **end for**
 7: **if** $\max_u(\lambda_{3,(a,u)}) > \tau_q$ **then**
 8:     **return** False
 9: **end if**
10: **return** True

---

---
**Algorithm 2** QUARTET: Coherence test
---
**Require:** quartet $(a, b, c, d)$, empirical moments $\hat{M}_{(a,b,c,d)}$, thresholds $\tau_q, \tau_q'$.
**Ensure:** boolean singly-coupled.
 1: **for** each triplet $(u, v, w)$ in $(a, b, c, d)$ **do**
 2:     $\theta_{(u,v,w)} \leftarrow \text{MIXTURE}((u, v, w))$
 3: **end for**
 4: **if** $\text{Variance}_{(u,v,w)}(\theta_{(u,v,w)}) > \tau_q$ **then**
 5:     **return** False
 6: **end if**
 7: $\theta_M \leftarrow \text{Mean}_{(u,v,w)}(\theta_{(u,v,w)})$
 8: $M_{\theta,(a,b,c,d)} = p(a, b, c, d; \theta_M)$
 9: **if** $|M_\theta - \hat{M}|_\infty > \tau_q'$ **then**
10:     **return** False
11: **end if**
12: **return** True

---

Figure 1: The two variants of the quartet test. **Left:** Rank test. **Right:** Coherence test. MIXTURE refers to using Eq. 2 to learn the parameters as though the triplet is singly-coupled.

## B   Additional experiments

To illustrate the nature of the noisy-or network, Figure 2 shows examples of training images drawn from the network.

We provide additional experiments to evaluate the sensitivity of the algorithm to the threshold parameters $\tau_q$ and $\tau_e$. Figure 3 shows the sensitivity of the algorithm to choices of the threshold variable $\tau_q$. We use the rank quartet test so $\tau_q'$ is not considered. As $\tau_q$ is decreased, fewer sources are discovered. When $\tau_q = 0.0001$ no sources are discovered at all. Increasing $\tau_q$ does not severely impact the performance of the algorithm since we sort the quartets by eigenvalue and choose the lower ones first (see description in Section 4), though the best performance is obtained with a manually optimized value of $\tau_q = 0.003$.

The sensitivity of the algorithm to choices of the threshold variable $\tau_e$ is shown in Figure 4. Setting $\tau_e$ too low results in learning a small number of sources that cover nearly the entire $8 \times 8$ grid.

Figure 2:  25 random samples from the synthetic distribution.  Each sample is displayed in a 8x8 pixel image.

Figure 3:  Sensitivity of the algorithm to threshold parameter $\tau_q$ with $\tau_e = 0.1$ learned with 10,000 samples.

After these sources are discovered, we are unable to find any new singly-coupled quartets.  When $\tau_e = 0.0001$, the first source discovered covers nearly all of the pixels, though most of them have failure rates close to 1.  Setting $\tau_e$ too high (i.e. $> 0.1$ not shown here), results in learning many sources each with a small number of children.

## C   Conditional point-wise mutual information

In this section, we restate the result of Lemma 2 and give its complete proof.

**Lemma.**  *Let $(a, b, x)$ be three observed variables in a noisy-or network, and let $\mathcal{U}_{a,b}$ be the set of common parents of $a$ and $b$. For $U \in \mathcal{U}_{a,b}$, defining*

$$p_{U|\bar{x}} = \frac{P(U, \bar{x})}{P(\bar{x})} = \frac{p_U f_{U,x}}{1 - p_U + p_U f_{U,x}},$$

*we have $p_{U|\bar{x}} \leq p_U$. Furthermore,*

$$\frac{P(\bar{a}, \bar{b}|\bar{x})}{P(\bar{a}|\bar{x})P(\bar{b}|\bar{x})} = \prod_{U \in \mathcal{U}_{a,b}} \frac{(1 - p_{U|\bar{x}} + p_{U|\bar{x}} f_{U,a} f_{U,b})}{(1 - p_{U|\bar{x}} + p_{U|\bar{x}} f_{U,a})(1 - p_{U|\bar{x}} + p_{U|\bar{x}} f_{U,b})} \leq \frac{P(\bar{a}, \bar{b})}{P(\bar{a})P(\bar{b})},$$

*with equality if and only if $(a, b, x)$ do not share a parent.*

*Proof.*  First we show that $\frac{P(\bar{a}, \bar{b})}{P(\bar{a})P(\bar{b})}$ and $\frac{P(\bar{a}, \bar{b}|\bar{x})}{P(\bar{a}|\bar{x})P(\bar{b}|\bar{x})}$ have a similar form, depending only on parents in $\mathcal{U}_{a,b}$ with $p_{U|\bar{x}}$ in the latter replacing $p_U$ in the former. Let $\mathcal{U}_{a,b}$ be the set of parents shared by $a$ and $b$, $\mathcal{U}_a$ be the set of parents of $a$ that do not have $b$ as a child, and $\mathcal{U}_b$ be the set of parents of $b$ that do not have $a$ as a child. Then $\mathcal{U} = \mathcal{U}_{a,b} \cup \mathcal{U}_a \cup \mathcal{U}_b$ is the set of parents of $a$ or $b$, and:

$$\frac{P(\bar{a}, \bar{b})}{P(\bar{a})P(\bar{b})} = \frac{\prod_{U \in \mathcal{U}}(1 - p_U + p_U f_{U,a} f_{U,b})}{\prod_{U \in \mathcal{U}}(1 - p_U + p_U f_{U,a}) \prod_{U \in \mathcal{U}}(1 - p_U + p_U f_{U,b})}$$

$$= \frac{\prod_{U \in \mathcal{U}_{a,b} \cup \mathcal{U}_a \cup \mathcal{U}_b} (1 - p_U + p_U f_{U,a} f_{U,b})}{\prod_{U \in \mathcal{U}_{a,b} \cup \mathcal{U}_a} (1 - p_U + p_U f_{U,a}) \prod_{U \in \mathcal{U}_{a,b} \cup \mathcal{U}_b} (1 - p_U + p_U f_{U,b})}$$

$$= \frac{\prod_{U \in \mathcal{U}_{a,b}} (1 - p_U + p_U f_{U,a} f_{U,b}) \prod_{U \in \mathcal{U}_a} (1 - p_U + p_U f_{U,a}) \prod_{U \in \mathcal{U}_b} (1 - p_U + p_U f_{U,b})}{\prod_{U \in \mathcal{U}_{a,b} \cup \mathcal{U}_a} (1 - p_U + p_U f_{U,a}) \prod_{U \in \mathcal{U}_{a,b} \cup \mathcal{U}_b} (1 - p_U + p_U f_{U,b})}$$

$$= \prod_{U \in \mathcal{U}_{a,b}} \frac{(1 - p_U + p_U f_{U,a} f_{U,b})}{(1 - p_U + p_U f_{U,a})(1 - p_U + p_U f_{U,b})}$$

Figure 4: Sensitivity of the algorithm to threshold parameter $\tau_e$ with $\tau_q = 0.01$ learned with 10,000 samples. On the left, edges are shown in grayscale according to their failure probabilities (white close to 0, black close to 1). On the right, all pixels that are children of a source are colored white to clearly show all of the edges. When $\tau_e$ is low, a small number of sources are learned that extend to nearly all of the observations.

Let $\mathcal{V}$ be the set of parents of any of $(a, b, x)$, defined similarly to $\mathcal{U}$ above. We then have:

$$P(\bar{a}, \bar{b}|\bar{x}) = \frac{P(\bar{a}, \bar{b}, \bar{x})}{P(\bar{x})}$$

$$= \prod_{U \in \mathcal{V}} \frac{(1 - p_U + p_U f_{U,a} f_{U,b} f_{U,x})}{(1 - p_U + p_U f_{U,x})}$$

$$= \prod_{U \in \mathcal{V}} (1 - \frac{p_U f_{U,x}}{1 - p_U + p_U f_{U,x}} + \frac{p_U f_{U,x}}{1 - p_U + p_U f_{U,x}} f_{U,a} f_{U,b})$$

$$= \prod_{U \in \mathcal{V}} (1 - p_{U|\bar{x}} + p_{U|\bar{x}} f_{U,a} f_{U,b}) = \prod_{U \in \mathcal{U}} (1 - p_{U|\bar{x}} + p_{U|\bar{x}} f_{U,a} f_{U,b})$$

The last line is true since $\mathcal{V} \setminus \mathcal{U}$ is the set of parents that are parents of $x$ but not parents of $a$ or $b$. The terms $(1 - p_{U|\bar{x}} + p_{U|\bar{x}} f_{U,a} f_{U,b}) = 1$ for all $U \in \mathcal{V} \setminus \mathcal{U}$ thus they can be ignored in the product.

Similarly, $P(\bar{a}|\bar{x}) = \prod_{U \in \mathcal{U}} (1 - p_{U|\bar{x}} + p_{U|\bar{x}} f_{U,a})$ and $P(\bar{b}|\bar{x}) = \prod_{U \in \mathcal{U}} (1 - p_{U|\bar{x}} + p_{U|\bar{x}} f_{U,b})$. Thus,

$$\frac{P(\bar{a}, \bar{b}|\bar{x})}{P(\bar{a}|\bar{x}) P(\bar{b}|\bar{x})} = \frac{\prod_{U \in \mathcal{U}} (1 - p_{U|\bar{x}} + p_{U|\bar{x}} f_{U,a} f_{U,b})}{\prod_{U \in \mathcal{U}} (1 - p_{U|\bar{x}} + p_{U|\bar{x}} f_{U,a}) \prod_{U \in \mathcal{U}} (1 - p_{U|\bar{x}} + p_{U|\bar{x}} f_{U,b})}$$

$$= \prod_{U \in \mathcal{U}_{a,b}} \frac{(1 - p_{U|\bar{x}} + p_{U|\bar{x}} f_{U,a} f_{U,b})}{(1 - p_{U|\bar{x}} + p_{U|\bar{x}} f_{U,a})(1 - p_{U|\bar{x}} + p_{U|\bar{x}} f_{U,b})}$$

Next we show that $p_{U|\bar{x}}$ has the following property:

$$p_{U|\bar{x}} = \frac{P(U, \bar{x})}{P(\bar{x})} = \frac{p_U f_{U,x}}{1 - p_U + p_U f_{U,x}} \leq \frac{p_U f_{U,x}}{(1 - p_U) f_{U,x} + p_U f_{U,x}}$$

$$= \frac{p_U f_{U,x}}{(1 - p_U + p_U) f_{U,x}} = p_U,$$

with $p_{U|\bar{x}} = p_U$ if and only if $f_{U,x} = 1$ (i.e. $x$ is not a child of $U$). Finally, the function:

$$g_{y,z} : \theta \mapsto \frac{1 - \theta + \theta yz}{(1 - \theta + \theta y)(1 - \theta + \theta z)}$$

is strictly increasing for $\theta \in [0, \frac{1}{1+\sqrt{yz}})$. $p_U$ is assumed to be less than $\frac{1}{2}$. Thus $g_{f_{U,a} f_{U,b}}(p_{U|\bar{x}}) \leq g_{f_{U,a} f_{U,b}}(p_U)$ with equality if and only if $x$ is not a child of $U$.

It follows that $\frac{P(\bar{a}, \bar{b}|\bar{x})}{P(\bar{a}|\bar{x}) P(\bar{b}|\bar{x})} \leq \frac{P(\bar{a}, \bar{b})}{P(\bar{a}) P(\bar{b})}$ with equality if and only if $a$, $b$ and $x$ have no parent that is shared by all three. $\qquad \square$

# D   Rank testability

Figure 5: Probability distribution of the third eigenvalue for a quartet with coupling parents, when all parameters are drawn uniformly at random. **Blue:** two parents, same failure probabilities. **Green:** three parents, same failure probabilities. **Red:** two parents, different failure probabilities. **Black:** three parents, different failure probabilities. This plot illustrates that rank testability only gets easier for random parameters compared to the uniform parameter setting used in the experiments.

**Rank test**.

Let $M$ be the $4 \times 4$ matrix representing the joint distribution of aggregated variables $(a, b)$ and $(c, d)$. Let $\mathcal{S}$ be the set of parents shared between two or more of those, and $\forall X \in \mathcal{S}, \forall u \in \{a, b, c, d\}$, let $f_{X,u}$ be the failure probability of the edge from $X$ to $u$, and let $n_u$ be the probability that $u$ is **not** activated by parents outside of $\mathcal{S}$.

$\forall S \subset \mathcal{S}, \forall u \in \{a, b, c, d\}$, let $e_{S,u} = n_u \prod_{X \in S} f_{X,u}$. $e_{S,u}$ is the marginal probability of $u$ being off given that nodes in $S$ are on and nodes in $\mathcal{S} \setminus S$ are off. We then have:

$$M = \sum_{S \subset \mathcal{S}} (\prod_{X \in S} p_X \prod_{Y \in \mathcal{S} \setminus S} (1 - p_Y)) q_S r_S^T$$

With:

$$U_S = \begin{pmatrix} e_{S,a} e_{S,b} \\ e_{S,a}(1 - e_{S,b}) \\ (1 - e_{S,a}) e_{S,b} \\ (1 - e_{S,a})(1 - e_{S,b}) \end{pmatrix}, V_S = \begin{pmatrix} e_{S,c} e_{S,d} \\ e_{S,c}(1 - e_{S,d}) \\ (1 - e_{S,c}) e_{S,d} \\ (1 - e_{S,c})(1 - e_{S,d}) \end{pmatrix}$$

In particular, this means that if $\{a, b, c, d\}$ only share one parent, the rank of $M$ is at most two (sum of two rank one matrices).

Conversely, if $|\mathcal{S}| > 1$, M is the sum of at least 4 rank 1 matrices, and its elements are polynomial expressions in the parameters of the model. The determinant itself is then a polynomial function of the parameters of the model $P(n_u, p_X, f_{X,u}; \forall u \in \{a, b, c, d\}, X \in \mathcal{S})$. Hence, if $P \not\equiv 0$, the set of its roots has measure 0. Table 1 gives two examples of parameter settings showing that $P \not\equiv 0$. For other structures, notice that these can also serve as examples by setting $p_U = 0$ for additional parents.

| $U$ | $p_U$ | $f_{U,a}$ | $f_{U,b}$ | $f_{U,c}$ | $f_{U,d}$ | $U$ | $p_U$ | $f_{U,a}$ | $f_{U,b}$ | $f_{U,c}$ | $f_{U,d}$ |
|---|---|---|---|---|---|---|---|---|---|---|---|
| $X$ | 0.2 | 0.2 | 0.4 | 0.6 | 1 | $X$ | 0.2 | 0.2 | 0.4 | 0.6 | 0.8 |
| $Y$ | 0.3 | 1 | 0.2 | 0.4 | 0.6 | $Y$ | 0.3 | 1 | 0.2 | 0.4 | 1 |

Table 1: Two settings where the determinant of the moments matrix is non zero **Left:** $Det(\theta_1) = 5.33 \times 10^{-7}$. **Right:** $Det(\theta_2) = 4.95 \times 10^{-7}$

Figure 6: Two networks with the same moments matrix. $p_X = 0.2$, $p_X = 0.3$, $p_Z = 0.37$. $f_X = (0.1, 0.2, 0.3)$, $f_Y = (0.6, 0.4, 0.5)$, $f_Z = (0.28, 0.23, 0.33)$. The noise and full moments matrix are given in Section E.

While these results show that rank testability holds with probability 1, they do not give much information on the more practical notion of $\epsilon$-rank testability. Figure 5 however provides an approximation of the probability density function of the third eigenvalue of the moments matrix of a noisy-or network whose parameters are drawn from a uniform distribution.

# E   Non identifiability of the structure.

Finding the latent structure of a network is an especially interesting problem. Not all network structures are identifiable. Indeed, by applying the tensor decomposition method to a triplet that shares two parents, we can often find one parent that would explain the moments just as well. Figure 6 gives an example of such a network. The parameters are the following:

```
original values
p = [0.20000000000000001, 0.29999999999999999]
f = [[0.10000000000000001, 0.20000000000000001, 0.29999999999999999],
 [0.59999999999999998, 0.40000000000000002, 0.5]]
n = [0.94999999999999996, 0.94999999999999996, 0.94999999999999996]

new
p = 0.369002801906
f = [0.27471379503828641, 0.22833778992716067, 0.3253928941874199]
n = [0.93603297899373428, 0.91486319364631141, 0.92461657113192608]

original_D [[[ 0.50557963   0.05459129]
  [ 0.06907822   0.05627086]]

 [[ 0.05137117   0.04281791]
  [ 0.06842098   0.15186994]]]

new_D [[[ 0.50557963   0.05459129]
  [ 0.06907822   0.05627086]]

 [[ 0.05137117   0.04281791]
  [ 0.06842098   0.15186994]]]

difference  = 3.46944695195e-17
```

# F   Sample complexity analysis

In this section, we give the full proofs of our sample complexity and correctness results. For simplicity we assume $f_{min} \leq 1 - f_{max}$. We split the theorem statement of Theorem 1 from Section 3 into two parts for ease of presentation.

**Theorem 1a.** *If a network with $m$ observed variables is strongly quartet-learnable and $\zeta$-rank-testable, then it can be learned with probability $(1 - \delta)$ with polynomial number of samples $N_S$:*

$$N_S = 3 \max(\frac{4 \times 2^{32}}{\zeta^8}, \frac{3 \times 1040^2}{n_{min}^8 p_{min}^2 (1 - f_{max})^8}) \ln(\frac{2m}{\delta})$$

**Theorem 1b.** *After $N$ samples, the additive error on any of the parameters $\epsilon(N)$ is bounded with probability $1 - \delta$ by:*

$$\epsilon(N) \leq \frac{5000 \times 1050 \times 11648 \times \sqrt{\ln\left(\frac{2m}{\delta}\right)}}{f_{min}^{18} (1 - f_{max})^6 n_{min}^{28} p_{min}^{13}} \frac{1}{\sqrt{N}}.$$

These results are proved in two steps. The structure learning algorithm is guaranteed to be correct if the estimation errors on the moments is below some thresholds. The first step of the proof consists of Lemmas 4 and 5 which give an expression of these thresholds as a function of the parameters of the model for the quartet and extending tests respectively:

**Lemma 4.** *If the model is $(\zeta)$-rank-testable and the estimation error on the fourth order moments of $(a, b, c, d)$ is bounded by $\epsilon < \zeta^4 / 2^{16}$, then the magnitude of the third eigenvalue of every $4 \times 4$ unfolding of the joint distribution is smaller than $\frac{\zeta}{2}$ if and only if $(a, b, c, d)$ is singly coupled.*

**Lemma 5.** *If the maximum estimation error $\epsilon$ on all of the moments up to third order is such that $1040\epsilon < n_{min}^4 p_{min} (1 - f_{max})^4$, $(a, b, x)$ share a parent if and only if:*

$$\frac{P(\bar{a}, \bar{b})}{P(\bar{a})P(\bar{b})} - \frac{P(\bar{a}, \bar{b}|\bar{x})}{P(\bar{a}|\bar{x})P(\bar{b}|\bar{x})} \geq \frac{p_{min}(1 - f_{max})^3 (1 - f_{max}^2)}{40}.$$

The second step characterizes how sensitive the parameter learning is to this same estimation error. The parameter learning algorithm first uses a tensor decomposition to find the prior of a new latent variable and the failure probabilities of the members of a singly-coupled quartet:

**Lemma 6.** *If the error on the moments is bounded by $\epsilon$, then the error on the parameters we obtain from the tensor decomposition of third-order moments is bounded by $\frac{5000 \times 1050\epsilon}{f_{min}^{18} p_{min}^{12} n_{min}^{24} (1 - f_{max})^3}$.*

The parameter learning algorithm then finds the failure probability for new children of a known latent variable using both the empirical moments and some previously estimated parameters. Hence, we need to consider how sensitive it is to the error on both these sets of values:

**Lemma 7.** *If the moments and parameters are known to within an error of $\epsilon$, the error on the failure probability we obtain from the extending step is bounded by $\frac{11648\epsilon}{(1 - f_{max})^3 n_{min}^4 p_{min}}$.*

Since the parameters used in the extending step are all given directly by the tensor decomposition, the final error on the parameters given the estimation error is then simply obtained by multiplying the factors of Lemmas 6 and 7.

Section F.1 presents some useful preliminary results. The correctness of the structure learning algorithm is proven in F.2 and the bound on the parameter estimation error is proven in section F.3.

## F.1 Preliminary Results

Throughout the proofs, we will need to be able to bound the error on a fraction and on the roots of a polynomial given the uncertainty on their terms. Lemmas 8 and 9 respectively give such bounds.

**Lemma 8.** *Let $\tilde{a} \in [a - \eta, a + \eta]$ and $\tilde{b} \in [b - \epsilon, b + \epsilon]$. If $\beta < b - \epsilon$ and $a + \eta < A$, then $\frac{\tilde{a}}{\tilde{b}} \in [\frac{a}{b} - (\frac{A}{\beta^2}\epsilon + \frac{1}{\beta}\eta), \frac{a}{b} + (\frac{A}{\beta^2}\epsilon + \frac{1}{\beta}\eta)]$*

**Lemma 9.** *Let $P(X) = aX^2 + bX + c$ and $\Delta = b^2 - 4ac$. Let $(x_1, x_2)$ be the roots of $P$, and suppose $(\tilde{a}, \tilde{b}, \tilde{c})$ are estimates of $(a, b, c)$ with an error bounded by $\epsilon$. Suppose $\exists d, 0 < d < \Delta$ and $\exists k > 0, -k < (\tilde{x_1}, \tilde{x_2}) < k$. Then the error on $(x_1, x_2)$ is bounded by $|(x_1, x_2) - (\tilde{x_1}, \tilde{x_2})| \leq \frac{2(1 + k + k^2)}{\sqrt{d}}\epsilon$.*

**Proof of Lemma 8.** We have,

$$\frac{a}{b+\epsilon} - \frac{\eta}{b-\epsilon} \le \frac{\tilde{a}}{\tilde{b}} \le \frac{a}{b-\epsilon} + \frac{\eta}{b-\epsilon}$$

$$\frac{a}{b} - \frac{a\epsilon}{b(b+\epsilon)} - \frac{\eta}{b-\epsilon} \le \frac{\tilde{a}}{\tilde{b}} \le \frac{a}{b} + \frac{a\epsilon}{b(b-\epsilon)} + \frac{\eta}{b-\epsilon}$$

$$\frac{a}{b} - \left(\frac{a\epsilon}{b(b-\epsilon)} + \frac{\eta}{b-\epsilon}\right) \le \frac{\tilde{a}}{\tilde{b}} \le \frac{a}{b} + \left(\frac{a\epsilon}{b(b-\epsilon)} + \frac{\eta}{b-\epsilon}\right)$$

$$\frac{a}{b} - \left(\frac{A\epsilon}{\beta^2} + \frac{\eta}{\beta}\right) \le \frac{\tilde{a}}{\tilde{b}} \le \frac{a}{b} + \left(\frac{A\epsilon}{\beta^s} + \frac{\eta}{\beta}\right)$$

**Proof of Lemma 9.** Let $K = 1 + k + k^2$, then $P(\tilde{x}_1) - K\epsilon < \tilde{P}(\tilde{x}_1) = 0 < P(\tilde{x}_1) + K\epsilon$, hence the estimated $\tilde{x}_1$ is between the roots of $P - K\epsilon$ and $P + K\epsilon$. Moreover, since $\sqrt{1+x} \in (1, 1+x)$:

$$\frac{b + \sqrt{b^2 - 4a(c+K\epsilon)}}{2a} = \frac{b + \sqrt{\Delta}\sqrt{1 - \frac{4aK}{\Delta}\epsilon}}{2a} \ge \frac{b + \sqrt{\Delta}}{2a} - \frac{2K}{\sqrt{\Delta}}\epsilon$$

$$\frac{b + \sqrt{b^2 - 4a(c-K\epsilon)}}{2a} = \frac{b + \sqrt{\Delta}\sqrt{1 + \frac{4aK}{\Delta}\epsilon}}{2a} \le \frac{b + \sqrt{\Delta}}{2a} + \frac{2K}{\sqrt{\Delta}}\epsilon$$

Hence $x_1 - \frac{2K}{\sqrt{\Delta}}\epsilon \le \tilde{x}_1 \le x_1 + \frac{2K}{\sqrt{\Delta}}\epsilon$. Similarly, $x_2 - \frac{2K}{\sqrt{\Delta}}\epsilon \le \tilde{x}_2 \le x_2 + \frac{2K}{\sqrt{\Delta}}\epsilon$.

### F.2 Correctness of the Structure Tests

At every stage of the algorithm, the moments are known to some error $\epsilon$. We first determine how small $\epsilon$ needs to be for the rank test to be guaranteed to successfully identify singly-coupled quartets:

**Lemma 4.** *If the model is $(\zeta)$-rank-testable and the estimation error on the fourth order moments of $(a, b, c, d)$ is bounded by $\epsilon < \zeta^4 / 2^{16}$, then the magnitude of the third eigenvalue of every $4 \times 4$ unfolding of joint distribution is smaller than $\frac{\zeta}{2}$ if and only if $(a, b, c, d)$ is singly coupled.*

**Proof of Lemma 4.** For each unfolding $M$, let $\tilde{M}$ be our estimate of $M$, and suppose $||M - \tilde{M}||_\infty \le \epsilon$, then the spectral norm of the difference $||M - \tilde{M}||_{sp} \le 4\epsilon$. We can then use a result from Elsner (1985) showing that, if $\lambda$ is the third eigenvalue of $M$ and $\tilde{\lambda}$ is the third eigenvalue of $\tilde{M}$, then:

$$|\lambda - \tilde{\lambda}| \le (||M||_{sp} + ||\tilde{M}||_{sp})^{\frac{3}{4}} ||M - \tilde{M}||_{sp}^{\frac{1}{4}} \le 8\epsilon^{\frac{1}{4}}.$$

Hence, if $\lambda \ge \zeta$, then $\tilde{\lambda} \ge \zeta - 8\epsilon^{\frac{1}{4}}$.

We then need to find how small the error on the empirical moments needs to be for the extending step to be correct:

**Lemma 5.** *If the maximum estimation error $\epsilon$ on all of the moments up to third order is such that $1040\epsilon < n_{min}^4 p_{min}(1 - f_{max})^4$, $(a, b, x)$ share a parent if and only if:*

$$\frac{P(\bar{a}, \bar{b})}{P(\bar{a})P(\bar{b})} - \frac{P(\bar{a}, \bar{b}|\bar{x})}{P(\bar{a}|\bar{x})P(\bar{b}|\bar{x})} \ge \frac{p_{min}(1 - f_{max})^3(1 - f_{max}^2)}{40}$$

**Proof of Lemma 5.** Indeed, we have:

$$\frac{P(\bar{a}, \bar{b})}{P(\bar{a})P(\bar{b})} - \frac{P(\bar{a}, \bar{b}|\bar{x})}{P(\bar{a}|\bar{x})P(\bar{b}|\bar{x})} = g_{f_{A,a}, f_{A,b}}(p_A) - g_{f_{A,a}, f_{A,b}}(p_{A|\bar{x}})$$

Moreover:

$$p_A - p_{A|\bar{x}} = p_A\left(1 - \frac{f_{A.x}}{1 - p_A + p_A f_{A.x}}\right)$$

$$\ge \frac{1}{2}p_{min}(1 - f_{max})$$

And, for $p \leq \frac{1}{2}$:

$$\frac{\partial g_{f_{A,a}, f_{A,b}}(p)}{\partial p} = \frac{(1 - f_{A,a})(1 - f_{A,b})((1 - f_{A,a}f_{A,b})p^2 - 2p + 1)}{(1 - p(1 - f_{A,a}))^2(1 - p(1 - f_{A,b}))^2}$$

$$\geq \frac{(1 - f_{max})^2(1 - f_{max}^2)}{8}$$

Hence:

$$\frac{P(\bar{a}, \bar{b})}{P(\bar{a})P(\bar{b})} - \frac{P(\bar{a}, \bar{b}|\bar{x})}{P(\bar{a}|\bar{x})P(\bar{b}|\bar{x})} > \frac{p_{min}(1 - f_{max})^3(1 - f_{max}^2)}{8}.$$

Suppose the estimation error on all of the probabilities is bounded by $\epsilon < \frac{n_{min}^2}{4}$, then the error on $P(\bar{a})P(\bar{b})$ is bounded by $3\epsilon$. We have $P(\bar{a})P(\bar{b}) > n_{min}^2$ and $P(\bar{a}, \bar{b}) \leq 1$, thus according to lemma 8 using $A = 1, \beta = n_{min}^2/4$, the estimation error on $\frac{P(\bar{a}, \bar{b})}{P(\bar{a})P(\bar{b})} - \frac{P(\bar{a}, \bar{b}|\bar{x})}{P(\bar{a}|\bar{x})P(\bar{b}|\bar{x})}$ is bounded by $2 \times (\frac{48}{n_{min}^4}\epsilon + \frac{4}{n_{min}^2}) \leq \frac{104}{n_{min}^4}\epsilon = \eta$.

Hence, if $\epsilon \leq \frac{n_{min}^4 p_{min}(1 - f_{max})^3(1 - f_{max}^2)}{104 \times 10}$, $\eta \leq \frac{p_{min}(1 - f_{max})^3(1 - f_{max}^2)}{10}$, which proves the lemma.

We can now prove the first part of our theorem.

**Proof of Theorem 1a.** According to Lemmas 4 and 5, we need to bound the error on fourth-order moments by $\eta_4$ and on the third-order moments by $\eta_3$ respectively. Using a Chernoff bound we get that after $N$ samples:

- if $N \geq \frac{3}{\eta_4^2} \ln(\frac{2m^4}{\delta})$, then the error on a fourth-order moment is smaller than $\eta_4$ with probability $(1 - \frac{\delta}{m^4})$, hence the maximum error on the fourth-order moments is bounded by $\eta_4$ with probability at least $(1 - \delta)$.

- Similarly, if $N \geq \frac{3}{\eta_3^2} \ln(\frac{2m^3}{\delta})$, then the maximum error on the third-order moments is bounded by $\eta_3$ with probability at least $(1 - \delta)$.

Taking the maximum of these numbers of samples, we find that with probability at least $(1 - \delta)$, all our structure tests are correct.

### F.3    Parameter Estimation Error

Even when the algorithm finds the right structure, the error in the parameters it returns depends on the difference between the true moments and the empirical moments. The magnitude of the error in the parameters learned by the tensor decomposition method is bounded by a multiplicative factor of this difference:

**Lemma 6.** *If the error on the moments is bounded by $\epsilon$, then the error on the parameters we obtain from the tensor decomposition of third-order moments is bounded by $\frac{5000 \times 1050}{f_{min}^{18} p_{min}^{12} n_{min}^{24}(1 - f_{max})^3}\epsilon$.*

**Proof of Lemma 6.** We seek a bound on the errors of $f_{X,u}$ and $p_X$ for a triplet $(u, v, w)$ singly coupled by latent variable $X$ as a function of $\epsilon$. Assume that matrices $X_1 = P(v, w, u = 0)$ and $X_2 = P(v, w, u = 1)$ are known to some element-wise error $\epsilon$. Let $Y_2 = X_2 X_1^{-1}$ and let $(\lambda_1, \lambda_2)$ be the eigenvalues of $Y_2$.
From Eq. 2 we have that:

$$f_{X,u} = \frac{1 + \lambda_1}{1 + \lambda_2}.$$

We first bound the error on $X_1^{-1}$. If $X_1 = \begin{pmatrix} a & b \\ c & d \end{pmatrix}$, then we have $X_1^{-1} = \frac{1}{ad - bc}\begin{pmatrix} d & -b \\ -c & a \end{pmatrix}$. Moreover, $(ad - bc) = p_X(1 - p_X)f_{X,u}(1 - f_{X,v})(1 - f_{X,w})n_1^2 n_2 n_3 \geq f_{min}^3 p_{min}^2 n_{min}^4$, so if $\epsilon < \frac{f_{min}^3 p_{min}^2 n_{min}^4}{2}$, we get using lemma 8 with $A = 1, \beta = \frac{f_{min}^3 p_{min}^2 n_{min}^4}{2}$ that the element-wise error on $X_1^{-1}$ is at most $\frac{25\epsilon}{f_{min}^6 p_{min}^4 n_{min}^8}$. Hence the error on the terms of $Y_2$ is bounded by $\frac{6 \times 25\epsilon}{f_{min}^6 p_{min}^4 n_{min}^8}$.

$(\lambda_1, \lambda_2)$ are the roots of the polynomial $P(X) = X^2 - Tr(Y_2)X + Det(Y_2)$ and $\Delta = |\lambda_1 - \lambda_2|^2 > (1 - f_{max})^2$. Moreover, $Det(Y_2) = \lambda_1\lambda_2 > 0$, so:

$$-\frac{2}{f_{min}^3 p_{min}^2 n_{min}^4} < -\frac{2}{ad - bc} < -|Tr(Y_2)| < (\lambda_1, \lambda_2) < |Tr(Y_2)| < \frac{2}{ad - bc} < \frac{2}{f_{min}^3 p_{min}^2 n_{min}^4}$$

Hence, according to Lemma 9, the error on the eigenvalues of $Y_2$ is bounded by $\frac{1050\epsilon}{f_{min}^{12} p_{min}^8 n_{min}^{16}(1 - f_{max})}$.

Furthermore, we have $n_1 = \frac{\lambda_1}{1 + \lambda_1}$, hence $1 + \lambda_1 = \frac{1}{1 - n} > 1$, hence if $\eta = \frac{1050\epsilon}{f_{min}^{12} p_{min}^8 n_{min}^{16}(1 - f_{max})} < \frac{1}{2}$, the error on $f_{X,u}$ is bounded by:

$$\frac{\eta}{1 + \lambda_1 - \eta} + \frac{(1 + \lambda_2)\eta}{(1 + \lambda_1)(1 + \lambda_1 - \eta)} \leq 2(1 + f_{max})\eta = \frac{2100(1 + f_{max})\epsilon}{f_{min}^{12} p_{min}^8 n_{min}^{16}(1 - f_{max})}$$

Let us now bound the error on the parameter $p_X$. According to Equation 2, we have:

$$p_X = \frac{1 + \lambda_2}{\lambda_2 - \lambda_1} \times \mathbf{1}^T(X_2 - \lambda_1 X_1)\mathbf{1} \ \ .$$

We showed that the error on $\lambda_1$ and $\lambda_2$ is bounded by $\eta$. Moreover, for $\epsilon \leq \frac{f_{min}^{12} p_{min}^8 n_{min}^{16}(1 - f_{max})}{8400}$, we can apply lemma 8 with $A = \frac{3}{f_{min}^3 p_{min}^2 n_{min}^4}$ and $\beta = \frac{1 - f_{max}}{4}$ to bound the error on $\frac{1 + \lambda_2}{\lambda_2 - \lambda_1}$ by $e_1 = \frac{100}{f_{min}^3 p_{min}^2 n_{min}^4(1 - f_{max})^2}$. Since the error on the individual terms of $X_1$ and $X_2$ is bounded by $\epsilon << \eta$, we can also coarsely bound the error on $\mathbf{1}^T(X_2 - \lambda_1 X_1)\mathbf{1}$ by $e_2 = 16 \times 2 \times \eta$.

Given our upper bound on $(\lambda_1, \lambda_2)$ and lower bound on $\lambda_2 - \lambda_1$, we also have $\frac{1 + \lambda_2}{\lambda_2 - \lambda_1} \leq m_1 = \frac{12}{f_{min}^3 p_{min}^2 n_{min}^4(1 - f_{max})}$, and $\mathbf{1}^T(X_2 - \lambda_1 X_1)\mathbf{1} \leq m_2 = \frac{48}{f_{min}^3 p_{min}^2 n_{min}^4}$. Hence, the error on $p_X$ is bounded by:

$$e_1 \times m_2 + e_2 \times m_1 + e_1 \times e_2 \leq \frac{5000 \times 1050\epsilon}{f_{min}^{18} p_{min}^{12} n_{min}^{24}(1 - f_{max})^3}.$$

We now bound the error introduced by the extending step, which takes as input both empirical moments and estimated parameters:

**Lemma 7.** *If the moments and parameters are known to within an error of $\epsilon$, the error on the failure probability we obtain from the extending step is bounded by $\frac{11648}{(1 - f_{max})^3 n_{min}^4 p_{min}}\epsilon$.*

**Proof of Lemma 7.** As stated above in the proof of Lemma 5, if the moments are known with error $\epsilon < \frac{n_{min}^2}{4}$, $R$ is known to within $\frac{52\epsilon}{n_{min}^4}$. The error on the coefficients of $Q(X)$ is then bounded by $7 \times \frac{52\epsilon}{n_{min}^4}$. Moreover, we know that one root is smaller than $\frac{1}{2}$ and that the other is bigger than $\frac{1}{1 + \sqrt{f_{A,a}f_{A,b}}}$, hence $\frac{\sqrt{\Delta}}{R(f_{A,a} - 1)(f_{A,b} - 1)} > \frac{1}{1 + \sqrt{f_{A,a}f_{A,b}}} - \frac{1}{2} > (1 - f_{max})$, which implies that $\Delta > (1 - f_{max})^6 R_{min}^2 > (1 - f_{max})^6$. We can then use lemma 9 with $k = 1$ to show that the error on the roots of $Q(X)$ is bounded by $\frac{7 \times 4 \times 52\epsilon}{(1 - f_{max})^3 n_{min}^4}$. Additionally assuming $\epsilon < p_{min}/2$ and using lemma 8 with $A = 1/2, \beta = 1/2$, the error on $f_{A,x}$ is then bounded by $\frac{1 - p_A}{p_A} \frac{2 \times 4 \times 1456\epsilon}{(1 - f_{max})^3 n_{min}^4} < \frac{11648\epsilon}{(1 - f_{max})^3 n_{min}^4 p_{min}}$.

We can now prove the second part of our theorem.

**Proof of Theorem 1b.** Using a Chernoff bound we have that the error in the empirical moments obtained from N samples is less than $\sqrt{\ln(\frac{2m}{\delta})}\frac{1}{\sqrt{N}}$ with probability $1 - \delta$. We combine the multiplicative factors on the error introduced by the tensor decomposition (Lemma 6) and during the extension step (Lemma 7) to achieve the stated result.

### F.4 Subtracting off

We proved that we could learn a class of **strongly** quartet-learnable noisy-or networks in polynomial time and sample complexity. We now give an extension of our algorithm to quartet-learnable networks.

The main idea behind the extension is the notion of subtracting off introduced in Section 1. Once we have learned all latent variables that have a singly coupled quartet of children, we can "remove" them from the network and obtain the moments that would have been generated by the rest of the network without them.

If some of the removed latent variables were coupling for otherwise singly coupled quartet, we can then discover new latent variables, and repeat the operation. If a network is quartet-learnable, we can find all of the latent variables in a finite number of subtracting off steps, which we call the **depth** of the network.

Since the tests for finding and extending latent variables are only guaranteed to work when the error in the moments is small, we need to check that this step does not introduce too big a difference between the empirical and the true subtracted off moments.

The terms of the joint distribution of 4 observed variables are obtained from the negative moments by inclusion/exclusion formulas, which have up to 15 terms, hence the error is at most 15 times the error on the negative moment, given by Lemma 3 of Section 3 which we restate and prove here:

**Lemma 3.** *If the additive error on the negative moments of an observed quartet $\mathcal{C}$ and on the parameters of the $l$ latent variables, $(X_1, \ldots, X_l)$, which we want to remove from $\mathcal{C}$ is bounded by $\epsilon$, then the error on the subtracted off moments is bounded by $2^{2l} \times 6(l+2)\epsilon$.*

**Proof of Lemma 3.** We have $p_{min} < p_X < \frac{1}{2}$ and $f_{min} < f_c \ \forall c \in \mathcal{C}$, so for $\epsilon < \frac{p_{min} f_{min}^4}{6}$, $(1 - \tilde{p}_X + \tilde{p}_X \prod_{c \in \mathcal{C}} \tilde{f}_{X,c}) > (1 - p_X + p_X \prod_{c \in \mathcal{C}} f_{X,u}) - 6\epsilon > \frac{1}{2}$. Moreover, if the error on $(1 - p_X + p_X \prod_{c \in \mathcal{C}} f_{X,c})$ is bounded by $6\epsilon$, then the error on $\prod_{j=1}^{l} (1 - p_{X_j} + p_{X_j} \prod_{c \in \mathcal{C}} f_{X_j,c})$ is bounded by $6(l+1)\epsilon$ for $\epsilon$ small enough ($\epsilon < \frac{1}{3l^2(1+\epsilon)^l}$), and $\prod_{j=1}^{l} (1 - \tilde{p}_{X_j} + \tilde{p}_{X_j} \prod_{c \in \mathcal{C}} \tilde{f}_{X_j,c}) > \left(\frac{1}{2}\right)^l$. Since the negative moment $N_{\mathcal{C}} < 1$, using Lemma 8, we then get that the error on the adjusted moment, $\tilde{N}'_{\mathcal{C}}$, is bounded by $2^{2l} \times 6(l+2)\epsilon$.

Using this lemma and the error propagation bound of Theorem 1b, we can prove a polynomial sample complexity for structure learning of quartet learnable noisy-or networks.

To that end, we introduce the **width** $W$ of the network, which is the maximum number of parents that need to be subtracted off to be able to learn the parameters for a new singly-coupled quartet. This leads to the following result, initially presented in Section 3:

**Theorem 2.** *If a network with $m$ observed variables is quartet-learnable at depth $d$, $\zeta$-rank-testable, and has width $W$, then its structure can be learned with probability $(1-\delta)$ with $N_S$ samples, where:*

$$N_S = O\left(\left(\frac{W 4^W}{f_{min}^{18}(1 - f_{max})^6 n_{min}^{28} p_{min}^{13}}\right)^{2d} \times \max\left(\frac{1}{\zeta^8}, \frac{1}{n_{min}^8 p_{min}^2 (1 - f_{max})^8}\right) \ln\left(\frac{2m}{\delta}\right)\right)$$

**Proof of Theorem 2.** The tests have the same requirements as in the proof for Theorem 1a, but each round of parameter learning introduces the multiplicative factor to the error given in Theorem 1b. Subtracting off after that introduces the multiplicative factor given in Lemma 3. Combining the three results proves the theorem.

## References

Elsner, Ludwig. 1985. An optimal bound for the spectral variation of two matrices. *Linear algebra and its applications*, **71**, 77–80.