[Reviews · NeurIPS 2013]

Submitted by Assigned_Reviewer_6

The authors present a method to learn the structure of a special class of Bayesian Noisy-Or networks (bipartite graphical models whose nodes are binary random variables).
To this end the authors search for special substructures among the observed variables, called quartets, using analysis of eigenvalues and existing results on tensor decompositions. They prove that their algorithm only depends polynomially on the number of samples and demonstrate the advantage in speed by comparing it on simulated data to an existing algorithm based on variational EM.

Clarity
I find the paper well written. The results are compared nicely to existing work and it has been made clear on which previous results their method depends. As a minor remark, I'd find the 'Background' section easier to parse with explicit formulas for the (conditional) probabilites.

The paper seems theoretically sound, though I did not check the proofs in the supplementary material.


Significance
Bayesian Noisy-Or networks can be used to model causal relations, where an effect is (probabilistically) caused by any event out of a given subset of mutually independent events. Therefore I think efficient methods of inferring the structure of these networks are important. Since exact inference has been shown to be computationally hard, structural assumptions on the decomposability/sparsity of the network are needed.
The paper proposes such an assumption and demonstrates that it is not too restrictive by showing that a large network of deseases (latent variables) and symptoms (observed variables) can be inferred almost completely (although, as I understand, deseases are modeled to be mutually independent, which may actually not be the case).


Originality
The results appear original to me (e.g. the combination of eigenvalue methods and tensor decomposition.)
Summary: The authors address the problem of inference of a Noisy-Or Bayesian network and present an efficient algorithm that can be applied to a subclass of these networks. This is a good quality paper that contains theoretical guarantees on the running time of the algorithm as well as demonstration of its performance in experiments.

Submitted by Assigned_Reviewer_9

This paper proposes a structure and parameter learning algorithm for noisy-or networks based on rank tests of fourth-order moments and spectral decomposition methods. The proposed algorithm builds on and generalizes existing algorithms for learning parameters of noisy-or networks when the structure is known. While the proposed method seems original and potentially useful, the overall presentation of the paper can be improved. More specifically,

(1) The background can be made more self-contained. Although the definitions of noisy-or Bayesian networks and negative moments can be inferred the description, it would be better if explicit definitions are given. Also, since the proposed work builds on Halpern and Sontag (2013), it would be nice to give a concrete, self-contained summary of that work. Without such a summary, the reader has nothing to refer to when encountering discussions of previous work, and may find the presentation hard to follow.

(2) In Algorithm 1, the "AdjustedJoint-Distribution" seems to be referring to the subtracting-off operation, while PRETEST was described in Section 5. They should be explicitly mentioned in, for example, the caption of Figure 2.

In addition, there are some technical questions:

(1) The tests and extension require threshold parameters. How sensitive are the algorithms to the values of these thresholds? How should one choose them?
Summary: The paper proposes a novel algorithm for learning structure and parameters of noisy-or Bayesian networks with theoretical guarantees. While the method seems original and potentially useful, the presentation of the paper needs to be improved.

Submitted by Assigned_Reviewer_10

The paper discusses parameter estimation and two versions of quartet-based structure learning algorithms for bipartite noisy-or networks of binary variables. The presented ideas are based on a recent paper [Halpern & Sontag, 2013], extending to a more general class of networks. The performance of the proposed algorithm is illustrated on synthetic examples. The exposition of paper is generally logical but the writing needs a lot of improvement.

The paper contains potentially interesting algorithms but it is at times hard to tell which ideas are new and which ones are borrowed. A summary of the main contributions and their significance would be useful. What are the real applications of the algorithm, for which existing algorithms would not be applicable?

* Main concerns:
-- Experiments: The authors list a number of practical applications but the performed experiments are rather limited.
-- Exposition: There are too many missing definitions, missing essential information (e.g., the algorithm by Halpern & Sontag ), inconsistent notation, and wrong pointers to other parts of the text. If this paper is accepted, I hope the authors take the task of improving the text seriously.

* Additional remarks:
-- Rank test: How practical is the epsilon-rank-testability assumption? In what applications would it be satisfied (with large epsilon)? Can epsilon be estimated from the data?
-- Definition 2: Please explain what you mean by "the third eigenvalue of ... is at least epsilon" if the eigenvalues are not necessary nonnegative real numbers.
-- Coherence test: For each quartet, 4 sets of parameters have to be estimated. State the computational cost of this estimation. "If the parameters learned do not agree" should be further explained.
-- The recent tensor decomposition method [Halpern & Sontag] seems to be an important part of the proposed algorithms and should at least briefly be explained.
-- The definition of noisy-or networks could be more thorough.
-- Some notation updates are required, e.g., on p.2, U stands for a set of latent variables but on p.3 U is used to describe a set of observed variables.
-- Figure 3 has wrong caption. Generally, Section 3 looks like it suffered last minute restructuring.
-- Figure 2 should come later, around the place it is referred to. The algorithms should probably not be called figures.
-- Thorough proofreading is necessary. There are multiple typos and missing references to figures and sections (in the main file and in the supplementary material).
Summary: This is a potentially interesting paper but the current version needs improvement. The main contributions and their significance should be made more clear. Further numerical experiments should be performed, preferably related to real applications. The presentation of the paper should be improved.
Author Feedback

Author rebuttal: We would like to thank the reviewers for their careful reading of the paper and will incorporate their suggestions to make the work more self-contained and clearer. We also recognize that some points could have been explained more thoroughly. We attempt to address these here and will integrate the clarifications into the revised version.

The main contributions of this work are the descriptions of the quartet and pointwise mutual information tests, along with their theoretical guarantees. While there are existing structure-learning algorithms that use rank-based quartet tests (we provide references in the “related work”), we are not aware of any that could apply to bipartite noisy-or networks, nor have we seen the pointwise mutual information test used to identify children of latent variables or estimate parameters.

The Halpern & Sontag paper uses tensor decomposition for parameter learning, but it is limited in that it requires that the network structure be known in advance. In our paper we show that it is possible to perform tensor-decomposition based factor analysis on datasets to learn bipartite noisy-or networks when the network structure, or even the number of latent variables, is not known in advance.

We use eigenvalues instead of singular values when describing the rank test in order to directly apply Elsner's result to prove the correctness of the algorithm with polynomial sample complexity. While this proof can be extended to singular values by bounding the L1 perturbation on a product of matrices, this requires an additional step and gives rise to a worse bound (although still polynomial). When we bound the eigenvalues away from 0, the bound applies to their magnitude.

Quartet learnability is difficult to determine without checking each network individually. However, some networks clearly do not fall into this category, and these may give some intuition about the concept. One such example would be if any pair of latent variables have close to identical sets of children (i.e. all children are shared except for at most two children of each parent). Intuitively, parents must have sufficiently non-overlapping sets of children in order for the network to be quartet learnable.

In the synthetic image dataset, the model is quartet-learnable, but not strongly. Following the order of the parents from Figure 3 of the supplementary material, latent variables 1 and 8 do not have singly coupled quartets without a subtraction step, but can be learned after subtraction. In Figure 4 of the main paper, for the models learned with 10K samples, the first 6 latent variables are learned without any subtraction and the rest are learned after subtracting off learned parents.

In the QMR-DT model, Table 1 shows that 469 latent variables are strongly quartet learnable, and that all but 4 are quartet learnable. If the structure is known, the number of latent variables that can be learned without a subtraction step is 527. When a latent variable is learned, its connections to all of its children are learned at the same time using pointwise mutual information.

--Epsilon-testability:
Epsilon-testability implies that it is hard to misidentify singly-coupled quartets as a result of sampling noise. If epsilon is large, that makes singly-coupled quartets easier to find. If epsilon is small, we require more samples in order to estimate the eigenvalues more accurately.

--- Implementation:
The pointwise mutual information R can be estimated from the empirical conditional negative moments using its definition from line 133 (also, l.h.s of line 145). All other quantities that go into the quadratic equation of line 161 have been previously learned.

--- Running time:
While we could have to check all O(n^4) quartets, which is costly (though still polynomial time), we filter candidate quartets by requiring that every pair of observations in the quartet have pointwise mutual information above a low threshold (we describe this process on line 393), reducing the number of quartets to 10% of its initial size.

The coherence test obtains four estimates of parameters through 4 tensor decompositions. Each decomposition is inexpensive, requiring solving a quadratic equation and the inversion and matrix multiplication of 2x2 matrices. The estimates "agree" if their variance is below a threshold.

--- Thresholds:
The threshold used by the quartet test determines the number of latent variables that are found; setting them too strictly results in not learning all of the latent variables, and too loosely results in discovering spurious latent variables. The threshold in the pointwise mutual information test determines the number of children ascribed to each latent variable. When it is too lax, each latent variable connects to many observations; when it is too stringent, each latent variable connects to very few observations.

To obtain our experimental results, we searched by orders of magnitude (10^k for k in [-6, -5, ..., 0]) until we found thresholds that gave reasonable looking results using the largest sample size (10,000 data points). We found that our results were not sensitive to the precise setting of the thresholds. However, thresholds that were orders of magnitude too high or too low did give clearly suboptimal results. In the absence of prior knowledge, one could use a model selection criterion such as BIC to choose between candidate thresholds.